# Effects of Plastic Film Mulching on Soil Enzyme Activities and Stoichiometry in Dryland Agroecosystems

**DOI:** 10.3390/plants11131748

**Published:** 2022-06-30

**Authors:** Meixia Liu, Xueqing Zhao, Md Elias Hossain, Shangwen Wang, Wenyi Dong, Subramaniam Gopalakrishnan, Enke Liu

**Affiliations:** 1Institute of Environment and Sustainable Development in Agriculture, Chinese Academy of Agricultural Sciences, Beijing 100081, China; 17854270793@163.com (M.L.); 82101215258@caas.cn (X.Z.); elias.abot@gmail.com (M.E.H.); wansunwen@163.com (S.W.); 2International Crops Research Institute for the Semi-Arid Tropics (ICRISAT), Patancheru 502324, India; 82101211155@caas.cn; 3State Key Laboratory of Hulless Barley and Yak Germplasm Resources and Genetic Improvement, Lhasa 850002, China

**Keywords:** nitrogen levels, maize of stage, enzyme activity, enzyme stoichiometric ratio, nutrient limitation

## Abstract

Soil extracellular enzymes are pivotal for microbial nutrient cycling in the ecosystem. In order to study the effects of different nitrogen application rates under plastic film mulching on soil extracellular enzyme activities and stoichiometry, five nitrogen application levels (i.e., 0, 90, 150, 225 and 300 kg·hm^−2^) were set based on two treatments: plastic film mulching (PM) and no film mulching (LD). We measured the soil extracellular enzyme activities (EEAs) and stoichiometry (EES) of four enzymes (i.e., β-1,4-glucosidase (βG), leucine aminopeptidase (LAP), β-1,4-N-acetylaminoglucosidase (NAG) and alkaline phosphatase (AP)) involved in the C, N and P cycles of soil microorganisms in surface soil at five maize growth stages (seedling stage, jointing stage, trumpet stage, grout stage and harvest stage). The results showed that there were significant differences in soil EEA at different maize growth stages. The soil nutrient content and soil EEA were significantly improved under PM, and the stoichiometric ratio of extracellular enzymes (E_C:N:P_) was closer to 1:1:1, which indicated that PM was beneficial to the balance of soil nutrients and the activity of microorganisms. At each stage, with the increase in nitrogen application levels, the soil EEA showed a trend of increasing first and then decreasing (or remained unchanged), and both LD and PM treatments reached their highest activity at the 225 kg·hm^−2^ nitrogen application rate. When the nitrogen application level was less than 225 kg·hm^−2^, the soil enzyme activity was mainly limited by the N nutrient, and when the nitrogen application level reached 300 kg·hm^−2^, it was mainly limited by the P nutrient. RDA and correlation analysis showed that the soil C:P, C:N, N:P and pH had significant effects on soil βG, NAG + LAP and AP activities as well as E_C:N_, E_C:P_ and E_N:P_.

## 1. Introduction

Soil extracellular enzymes are secreted by microorganisms and participate in the material circulation and energy flow of ecosystems [1]. Because of their functional characteristics and sensitivity to environmental changes, soil extracellular enzymes (EEAs) are powerful indicators of the soil quality and microbial metabolic activity [2,3]. A large number of studies have shown that four enzymes (i.e., β-1,4-glucosidase (βG), leucine aminopeptidase (LAP), β-1,4-N-acetylaminoglucosidase (NAG) and alkaline phosphatase (AP)) can be combined with the C, N and P cycles of microorganisms [1,4]. Soil extracellular enzyme stoichiometry (EES) is calculated from the ratio of C-, N- and P-acquiring enzymes (i.e., βG, NAG + LAP and AP), and it represents the relative abundance of different functional enzymes [1]. By evaluating the change in microbial metabolism from nutrient flow to energy flow, we can gain insight into the soil energy and nutrient limitation. Sinssabaug et al., 2009 discovered that the quantitative ratio of C:N:P enzyme activity in topsoil on a global scale, namely, ln(βG):ln (NAG + LAP):ln (AP), was 1:1:1 [1]. When the ratio deviated from 1:1:1, microorganisms were considered to be limited by C, N or P [5]. Others further calculated the vector length (VL) and angle (VA) of the stoichiometric ratio of soil enzymes to reflect the restriction of soil microorganisms [6]. The larger the VL value, the more severe the restriction of microorganisms relative to C; VA < 45°and VA > 45°, respectively, indicate the N-uptake limit and P-uptake limit of microorganisms [6,7]. Peng et al., 2016 pointed out that the contents of total carbon, total nitrogen and total phosphorus in 0~10 cm surface soil had a great influence on the soil EEA and EES, indicating that the soil EES was largely controlled by soil nutrient stoichiometry [8].

The Chinese Loess Plateau is sensitive to anthropogenic disturbances and climate change [9]. Because plastic film mulching has good heat preservation and water retention effects, it plays an important role in promoting crop yields, and it is widely used in northern China [10,11]. According to the statistics of the Food and Agriculture Organization of the United Nations, approximately 50% of the increase in world grain output is attributed to the application of chemical fertilizers, among which the contribution of chemical nitrogen fertilizer is the most significant [12], but the utilization rate of nitrogen fertilizer is only approximately 30% in the current season [13].

The published literature has focused on how different levels of added nitrogen affect EEA. After eight years of experiments, Li et al., 2021 found that nitrogen addition had a significant impact on the soil EEA, which was related to the form and levels of nitrogen addition [14]. Chen et al., 2016 conducted a meta-analysis of 62 studies and showed that nitrogen addition significantly improved the soil enzyme activity [15]. Studies have also proved that the excessive application of nitrogen fertilizer can easily lead to soil nitrogen saturation and aggravate soil microbial phosphorus limitation [16]. However, currently, few studies have been carried out on the response mechanism of soil EEA during the whole growth period under the comparison of film mulching and no film mulching treatments. There is insufficient information on the effect of different nitrogen application levels on soil EEA with film mulching. In the dryland farmland ecosystem in northern China, the soil EEA is mainly affected by artificial measures such as fertilization [17]. Studying the soil EEA and its stoichiometric characteristics at different nitrogen application levels during maize growth will help us to better understand the heterogeneity of ecological stoichiometric characteristics at different time scales between different nitrogen application levels under plastic film mulching. In addition, changes in ecological stoichiometry may reveal how biogeochemical cycles promote crop growth, which is important for guiding fertilization and yield increase.

This study carried out an investigation on the characteristics of soil enzyme activity and the stoichiometric characteristics of spring maize at different nitrogen application rates under plastic film mulching in northern China in order to provide some scientific basis for the sustainability of the soil micro-ecological environment of regional plastic film mulching cultivation. The purpose of this study was to explore the following questions: (1) Are there significant differences in the soil EEA and EES under PM and LD treatments? Does it change with the growth period of maize? (2) What is the change rule of soil EEA and EES under different nitrogen application levels? (3) What are the main environmental factors that affect soil EEA and EES? We hypothesized that: (1) the soil EEA and EES could be significantly different under PM and LD treatments and change in different growth periods of maize; (2) soil EEA and EES can change with the nitrogen application levels; (3) compared with the content of soil nutrients, the ratio of soil nutrients (i.e., C:N, C:P and N:P) could have a greater impact on EEA and EES in surface soil.

## 2. Results

### 2.1. Physical and Chemical Properties of Soil

The contents of pH, SOC, TN, and TP in soil showed significant differences with the growth period of maize (*p* < 0.05) (Figure 1). On the whole, the soil nutrient content showed the trend of PM < LD, and the correlation analysis showed that there was a significant difference between treatments (*p* < 0.05) (Figure 1). There was no significant difference between PM and LD treatments on soil pH at the seedling stage and jointing stage (*p* < 0.05). The contents of SOC, TN and TP in soil increased at first and then decreased with the increase in nitrogen application levels, reaching the highest when nitrogen application levels were 225 kg·hm^−2^ on the whole. Correlation analysis showed that there were significant differences among different nitrogen application levels (*p* < 0.05) (Figure 1). The results of multifactor correlation analysis showed that film mulching and nitrogen application had an effect on the SOC content at the seedling stage and harvest stage. The soil TN content at the seedling stage, jointing stage, trumpet stage, grout stage and harvest stage; soil TP content at the seedling stage, jointing stage, trumpet stage and grout stage showed significant interaction (Figure 1).

### 2.2. Soil Extracellular Enzyme Activities and Its Stoichiometry

Except for the seedling stage, the soil EEAs under the PM and LD treatments were the highest compared to the other four periods when the nitrogen application levels were 225 kg·hm^−2^. At the seedling stage, the activity of NAG + LAP gradually decreased with the increase in nitrogen application levels (Figure 2). Compared with the LD treatment, the EEA under the PM treatment was higher than that under LD (Figure 2). Multivariate ANOVA showed that there was an obvious interaction between plastic film mulching and nitrogen application levels (Figure 2).

Under the LD treatment, the E_C:N_, E_C:P_ and E_N:P_ values were in the range of 0.52–0.88, 0.61–0.86 and 0.82–1.17, respectively. Under the PM treatment, the E_C:N_, E_C:P_ and E_N:P_ values were in the range of 0.49–0.97, 0.54–0.89 and 0.80–1.14, respectively. The values of E_C:N_ and E_C:P_ in the soil in this experiment were all less than one (Figure 3), which indicates that soil microorganisms are limited by N and P nutrients at the same time (Figure 3 and Figure 4). We found that with the increase in nitrogen application levels, the soil E_C:N_ value gradually increased, approaching one (Figure 3). Under the LD treatment, the soil E_C:P_ showed a trend of increasing first and then decreasing with increase in nitrogen application levels. At the 225 kg·hm^−2^ nitrogen application level, the soil EEA was the highest. Under the PM treatment, the soil E_C:P_ ratio remained unchanged when the nitrogen application levels were 150, 225 and 300 kg·hm^−2^. At the same time, we found that the E_C:N_ and E_C:P_ in the soil were the closest to one at the jointing stage, and E_C:N_ and E_C:P_ ratios of soil microorganisms remained basically unchanged at the trumpet stage, grout stage and harvest stage, showing the rule of PM > LD as a whole (Figure 3). During the harvest period, when the nitrogen application levels were 0, 90, 150 and 225 kg·hm^−2^, the soil E_N:P_ value was generally greater than one, especially in the seedling stage, jointing stage and trumpet stage (Figure 3, Figure 4 and Figure 5). When the nitrogen application level was 300 kg·hm^−2^, the soil VA < 45° (Figure 5). Compared with LD treatment, the soil E_N:P_ ratio of PM was closer to one at the seedling stage, jointing stage and grout stage, indicating that PM is more conducive to the balance of soil nutrients.

The soil E_C:N:P_ of the LD and PM treatments were 0.73:1:1.16 and 0.79:1:1.03, respectively. Two-way ANOVA showed that PM measurements and nitrogen application levels significantly affected the EEA, and there was a significant interaction between them.

### 2.3. Relationship between Soil Physical and Chemical Properties and Soil Enzyme Activity and Its Stoichiometric Ratio

The RDA analysis (Figure 6) shows that the first axis explains 69.2% and 63.5% of the variation in enzyme activity and enzyme stoichiometric ratio under the LD and PM treatments, and the second axis explains 4.4% and 4.6%, respectively. The difference in soil N:P was 51.7% and 4.7%, respectively. Soil pH accounted for 14.5% and 37.4% of the difference, respectively. At the same time, C:N, SOC and TP explained 2.2%, 1.8% and 2.4% of the LD treatment, respectively, and C:P and TN explained 20.9% and 2.5% of the PM treatment, respectively. The correlation analysis (Figure 6) shows that soil pH was positively correlated with βG and AP (*p* < 0.01); the SOC content in the soil showed a highly significant positive correlation (*p* < 0.01) with E_C:N_ and E_C:P_ and a significant negative correlation (*p* < 0.05) with E_C:N_. The C:N content in soil showed a significant positive correlation with soil βG under LD and PM treatments (*p* < 0.05), and a very significant positive correlation with AP under LD treatment (*p* < 0.01). The contents of C:P and C:N in soil showed a highly significant positive correlation (*p* < 0.01) with soil E_C:N_ and E_C:P_ under the LD and PM treatments, and a highly significant negative correlation (*p* < 0.01) with E_N:P_ (Figure 6).

## 3. Discussion

Soil enzymes, as participants of material circulation and energy transformation in the soil ecosystem, are closely related to soil environmental conditions and are widely used to evaluate soil quality and soil biochemical activity [18]. Soil βG, NAG, LAP and AP are sensitive to soil environment changes and play an important role in the cyclic transformation of soil C, N and P [1,19]. Because PM can change the soil water and soil nutrients, it can affect soil enzyme activities by influencing the soil microbial decomposition and mineralization of soil organic matter, plant growth and metabolism level, and other factors [8,17,20].

### 3.1. Effects of Different Nitrogen Application Levels on Soil Physical and Chemical Properties under PM Treatments

Plastic film mulching can improve soil physical and chemical properties by improving the soil microenvironment [21,22,23]. Our results showed that the soil nutrient content in each growth stage of maize was higher under the PM treatment than under the LD treatment on the whole, indicating that plastic film mulching played an important role in improving the soil nutrient content, especially at the seedling and jointing stages (Figure 1). Relevant studies have shown that long-term plastic film mulching can influence the availability of soil nutrients [24]. Meanwhile, the SOC, TN and TP contents under the PM and LD treatments showed an increasing trend with the increase of in nitrogen application levels and reached the highest level at 225 kg·hm^−2^ on the whole (Figure 1, Table 1). Other studies have shown that nitrogen addition to the activity of C-cycling enzymes may lead to instability in the soil C pool [14] and that excessive nitrogen fertilizer input could promote the fixation of P nutrient, limiting the P availability in soil [25]. It can be seen that the proper amount of nitrogen fertilizer can significantly increase the soil nutrient content, increase the soil available nutrients and effectively promote the growth of aboveground crops. Excessive nitrogen application can change the fertilizer supply capacity of soil by affecting the composition and quantity of soil organic carbon, nitrogen and phosphorus.

### 3.2. Effects of Different Nitrogen Application Levels on Soil EEA under PM Treatments

Our results showed that the activities of soil enzymes involved in C, N and P cycles in farmland soil were significantly increased under plastic film mulching treatment as a whole, and the E_C:N:P_ ratio of soil was closer to 1:1:1, indicating that the supply level and health level of soil C, N and P were significantly improved after plastic film mulching. Relevant research shows that film mulching significantly affects the production of extracellular enzymes in soil [20]. At the same time, there were significant differences in the soil nutrient content and soil enzyme activity among different nitrogen application levels in this experiment (*p* < 0.05). Regarding the activity of βG and AP in soil, the soil EEA showed a trend of in increasing first and then decreasing with the increase in nitrogen application levels. Overall, the soil enzyme activity at the 225 kg·hm^−2^ nitrogen application level was significantly higher than that of other nitrogen application levels under PM and LD treatments. Li et al., 2021 reported that the appropriate nitrogen application level can significantly improve the soil nutrient content and soil enzyme activities [14]. Other studies found that nitrogen addition increased the soil AP activity, revealing that higher nitrogen availability stimulated the biological demand for phosphorus [26,27]. Therefore, different nitrogen application levels could have a significant impact on the concentration of soil enzyme activity in surface soil. Reasonable nitrogen application levels can improve soil fertility, increase soil available nutrients, promote crop growth and be more conducive to improving soil enzyme activity.

In our study, we found that the activities of soil enzymes involved in soil C, N and P cycles were generally low at the seedling stage. There could be two possible reasons for this phenomenon: first, the chemical fertilizer applied in the soil during the sowing time is still sufficient in the soil at this time, resulting in a low EEA in the soil; second, in the early stage of maize growth, the low temperature in northern China inhibited soil microbial activity and crop root activity, which may also lead to the low EEA. After the jointing stage, the soil enzyme activity showed an upward trend as a whole. After the trumpet stage, the soil βG activity remained basically unchanged and the soil NAG + LAP activity gradually decreased, while the AP activity still maintained an upward trend and the soil enzyme activity under plastic film mulching treatment was significantly higher than that under no film mulching treatment. The activity of NAG + LAP in soil decreased significantly as the nitrogen level increased at the seedling stage. This is consistent with the research results of Sinsabaugh et al., 2016, which showed that the difficulty of obtaining N by microorganisms is effectively reduced with the increase in the nitrogen application level [28]. Therefore, at this time, the soil enzyme activity was low under the high nitrogen application levels. However, after the jointing stage, especially in the trumpet stage and grout stage, soil NAG + LAP enzyme activity showed a trend of increasing first and then decreasing with the increase in nitrogen application levels. This is because since the jointing stage, the temperature in northern China has gradually increased, the rainfall has increased, the maize has grown vigorously, the root activity has increased, the secretion has increased, and the number of soil microorganisms has increased, so that the soil enzyme activity has increased significantly [29].

### 3.3. Effects of Different Nitrogen Application Levels on Soil EES under Plastic Film Mulching Treatment

The ratio of soil E_C:N:P_ is approximately 1:1:1 at the global level. However, its value will also change in different regions, vegetation types, and land conditions [14]. When the ratio deviates from 1:1:1, this indicates that soil microorganisms are limited by carbon, nitrogen or phosphorus [1,26]. The results showed that the stoichiometric ratio of soil extracellular enzymes showed significant differences among the treatments (Figure 3).

On the whole, PM measures increased soil E_C:N_ and E_C:P_ values, and decreased E_N:P_ values. In this experiment, the values of E_C:N_ and E_C:P_ in the soil were all less than one, which indicates that the soil microorganisms were limited by both N and P nutrients. Studies have shown that the content of total N and P in farmland soil is low in the Loess Plateau, which directly leads to the restriction of nutrients on microorganisms [30]. With the increase in nitrogen application levels, the E_C:N_ value in soil gradually increased, approaching 1 under the PM and LD treatments. However, the soil E_C:P_ value showed a trend of increasing first and then decreasing with the increase in nitrogen application levels in LD treatment, reaching the maximum when the nitrogen application level was 225 kg·hm^−2^ on the whole. After applying 150 kg·hm^−2^ fertilizer, the soil E_C:P_ value in the PM treatment did not change with the nitrogen application rate. This indicated that both plastic film mulching application and nitrogen fertilizer levels could effectively improve soil enzyme activities, especially when the nitrogen application level was 225 kg·hm^−2^. This showed that when soil nutrients are sufficient, there could be a higher soil enzyme activity.

The results show that compared with phosphorus, the soil at this stage is mainly limited by nitrogen when the nitrogen application levels is lower than 225 kg·hm^−2^. In the 300 kg·hm^−2^ nitrogen application level, the value of E_N:P_ in the soil was less than one, and VA was less than 45°, indicating that the soil was mainly limited by P nutrients. Relevant research shows that soil microorganisms can adjust their physiological metabolism by changing the expression of enzymes, so as to adapt to the changes in the external environment [13]. Previous studies have shown that nitrogen addition promotes plant growth, intensifies the demand of soil microorganisms for phosphorus, accelerates the absorption of phosphorus in soil and reduces the content of phosphorus in soil [14]. Compared with LD treatment, we found that the soil E_N:P_ of PM treatment was closer to one at the seedling stage, jointing stage and grout stage. It further showed that PM measures are more conducive to the balance of soil nutrients.

### 3.4. Soil Environmental Factors Related to Soil EEA and EES

A large number of studies have shown that the soil pH, water content and nutrient availability will affect the soil enzyme activity to some extent [15,18]. Our results showed that the contents of C, N and P in the soil and the influence degree of the stoichiometric ratio on soil enzyme activity and the enzyme stoichiometric ratio were different, which directly affected the quality and quantity of the soil enzyme activity. Studies have shown that the increase in pH value is not conducive to the release of phosphorus in soil, thus reducing the content of inorganic phosphorus in soil and promoting the activity of alkaline phosphatase in soil. This is consistent with this research’s conclusion [25]. At the same time, under the condition of nitrogen addition, soil TN and AP activities in PM and LD treatments showed a negative correlation and a positive correlation with E_C:P_, indicating that nitrogen addition aggravated the phosphorus limitation of soil microorganisms. With the increase in the nitrogen application level, the demand for TP will increase, and it will be more difficult for soil microorganisms to obtain phosphorus, which will stimulate the expression of serine or threonine protein phosphatase *PHO* gene in soil, thus promoting the microbial synthesis of AP [27]. C:N showed a significant negative correlation with soil enzyme activity under LD treatment. However, the SOC content showed a significant positive correlation with soil enzyme activity under film mulching treatment. There is a negative correlation between N:P and the soil enzyme activity under film mulching treatment, which indicates that the soil enzyme activity is influenced by many factors. Relevant studies have pointed out that in 0–10 cm topsoil, the soil total nitrogen, total carbon and total phosphorus content have the greatest influence on soil extracellular enzyme activity and soil enzyme chemometrics [6]. The content of soil nutrients affects the soil extracellular enzyme activity and enzyme chemometrics through the concentration of soil effective mechanisms and the ratio of chemical forces of C, N and P in the soil.

## 4. Materials and Methods

### 4.1. Site Description

The research was carried out at the Shouyang key field scientific observation experimental station (37°44′52″ N, 113°12′11″ E), Ministry of Agriculture and Rural Affairs, Shouyang County, Jinzhong City, eastern Shanxi Province. This area is a typical dry farming area in eastern Shanxi and western Henan provinces, with an altitude of 1202 m and a temperate continental monsoon climate zone of the semi-humid and arid climate type; the experimental land is flat with no irrigation conditions. The annual average temperature in this area is 7.4 °C, the frost-free period is approximately 140 days, and the sunshine hours are 2858.3. The average annual precipitation in the last five years was 518.3 mm, which was unevenly distributed during the year, mainly from June to September, accounting for more than 80% of the annual rainfall.

### 4.2. Experimental Design and Treatments

This experiment began in 2015. The planting system was a one-year cropping system, and the planting crop was spring maize. Two treatments, plastic film mulching (PM) and no film mulching (LD), were set in the experiment. Each treatment was set with five different nitrogen application levels of 0, 90, 150, 225 and 300 kg·hm^−2^, and each treatment had three repetitions, with a total of 30 plots, each of which was 48 m^2^ (length 8 m × width 6 m), and a completely random block design was adopted. The spring maize variety “Zhengdan 958” had a planting density of 60,000 plants·hm^−2^, a row spacing of 50 cm and a plant spacing of 33.3 cm. The amount of fertilizer used in each treatment was the same, with urea (46.7% N) 240 kg·hm^−2^ as nitrogen fertilizer, superphosphate (16% P_2_O_5_) 150 kg·hm^−2^ as phosphorus fertilizer, and potassium chloride (60% KCl) 75 kg·hm^−2^ as potassium fertilizer. The plastic film was laid manually, and the plastic film was a slight white ordinary PE agricultural plastic film with a thickness of 0.008 mm and a width of 1.2 m.

Soil samples with a depth of 0~20 cm were collected in five periods in 2021: the maize seedling stage, jointing stage, trumpet stage, grout stage and harvest stage. According to the “V” sampling method, a spiral drilling needle with a diameter of 5 cm was used for three rounds of repeated drilling. Parts of soil samples were screened with a 2 mm sieve, put into self-sealing bags, transported back to the laboratory with ice bags, and stored at 4 °C for the determination of soil extracellular enzyme activity. The rest of the soil samples were air-dried and screened with 2 mm and 100 mesh screens, respectively, which were used to determine the soil pH, soil organic carbon (SOC), total nitrogen (TN), and total phosphorus (TP).

### 4.3. Soil Physiochemical Analysis

The pH of the soil was measured by the water immersion potential method, and the water–soil ratio was 5:1.

SOC was measured by the potassium dichromate external heating method: weigh 0.1000 g air-dried soil of 100 meshes in a digestion tube; add 5 mL of potassium dichromate, then 5 mL of concentrated sulfuric acid; cover with a small funnel, put it in an oil bath at 170–180 °C, and boil for 5 min; after cooling, transfer approximately 60–70 mL of the liquid to a triangular bottle. Add 5–8 drops of indicator, titrate with 0.2 mol/L FeSO_4_, and record the consumption.

The soil TN was determined by the semi-micro Kjeldahl nitrogen determination method (KDN-9830): 0.6 g air-dried soil of 100 meshes was weighed in a digestion tube, 3 g accelerator and 8 mL concentrated sulfuric acid were added for digestion, all the digestion liquid was transferred to a distillation tube, and the Kjeldahl nitrogen determination instrument was used for determination.

The soil TP in soil was measured by molybdenum antimony colorimetry (752N): 0.6 g air-dried soil of 100 meshes was weighed in a digestion tube, and then 8 mL concentrated sulfuric acid and 10 mL perchloric acid were added for digestion, and all the digestion solution was transferred to a 100 mL volumetric flask for constant volume. A measure of 10 mL was sucked into a 50 mL volumetric flask and two drops of 2,4-dinitrophenol were added. Alkali was used to make it yellow, and the acid was diluted to make it colorless. Then, 5 mL molybdenum antimony reagent was added for colorimetric determination; water was added to make the volume 50 mL, and then the mixture was shaken well. The mixture was left for 30 min and was used to compare the color at a 700 nm wavelength. Refer to soil agrochemical analysis for specific methods [31].

### 4.4. Soil Extracellular Enzyme Activity and Enzyme Stoichiometry Ratio

The activity of β-1,4-glucosidase (βG) closely related to the C cycle, β-1,4-N-acetylaminoglucosidase (NAG) and leucine aminopeptidase (LAP) closely related to the N cycle, and alkaline phosphatase (AP) closely related to the P cycle were determined by modified microplate fluorescence method [32].

The extraction solution configuration: acetic acid buffer (c (CH_3_COONa) = 50 mmol L^−1^): 4.1 g analytically pure anhydrous sodium acetate, dissolved in 1 L water. The pH was adjusted with acetic acid or sodium hydroxide according to the soil sample to be measured.

Preparation of the substrate: β-1,4-glucosidase (c (βG) = 2000 μmol·L^−1^): accurately weigh 0.0169 g of substrate in a 25 mL brown volumetric flask. leucine aminopeptidase (c (LAP) = 2000 μmol·L^−1^): accurately weigh 0.0162 g of substrate in a 25 mL brown volumetric flask. β-1,4-N-acetylaminoglucosidase (c (NAG) = 2000 μmol·L^−1^): accurately weigh 0.0190 g of substrate in a 25 mL brown volumetric flask. Alkaline phosphatase (c (AP) = 2000μmol·L^−1^): accurately weigh 0.0128 g of substrate in a 25 mL brown volumetric flask.

Preparation of standard curve: accurately weigh 0.0088 g of MUB (4-methylumbelliferone) into a 25 mL brown volumetric flask to obtain 2000 μmol·L^−1^ MUB and standard solution, and then suck 10 mL of the standard solution to 100 mL to obtain 200 μmol·L^−1^MUB standard solution. Accurately absorb 0, 1.25, 2.5, 5, 12.5, 25 and 37.5 mL of 200 μmol·L^−1^MUB standard solution to 50 mL and obtain the standard gradients of MUB concentrations of 0, 5, 10, 20, 50, 100, 150 and 200 μmol·L^−1^, respectively.

Determination step: weigh 2.75 g of fresh soil sample, add 91 mL of 50 mmol·L^−1^ sodium acetate buffer solution, and shake at 200–220 rpm·min^−1^ for 30 min to obtain soil suspension. Then, add 200 μL of the suspension and 50 μL of the corresponding substrate to the 96-well ELISA plate to make sample micropores: eight replicates for each sample. All samples were cultured in the dark at 25 °C for three hours. After the culture, the samples were excited at 365 nm with a microporous multifunctional microplate reader, and the fluorescence of the samples was tested at 450 nm. At last, the activity of soil extracellular enzymes was calculated by the dry weight of soil and the reaction time, which was expressed as nmol·activity·g^−1^·dry·soil·h^−1^.

Calculation method: Calculate the amount of MUB released from the sample plate according to the regression equation calculated by the marking line, then calculate the relative enzyme activity in each well according to the amount of MUB released and subtract the blank control from the enzyme activity of the soil sample to obtain the final soil enzyme activity.
MUB_soil_ = (F_soil_ − b)/a
E_a_ = (MUB_soil_ × ts × 1000):(t × m)
E_soil_ = E_a_ − E _bulk_

Among them, “MUB_soil_” is the amount of MUB released by the reaction between soil sample and substrate; “F_soil_” is fluorescence data, measured by the sample plate; “b” is the intercept of the marking line; “a” is the slope of the marking line. “ts” is the fractional multiple; “1000” is the conversion of μmol to nmol; “t” is the culture time; “m” is the drying quality of soil (nmol·activity·g^−1^·dry·soil·h^−1^).

Microbial C:N, C:P and N:P acquisition (E_C:N_, E_C:P_, and E_N:P_, respectively)
E_C:N_ = lnβG:ln(NAG + LAP)
E_C:P_ = lnβG:ln(AP)
E_N:P_ = ln(NAG + LAP):lnAP

The vector length (VL) and vector angle (VA) of enzyme measurement were used to detect the relative limitation of energy and nutrition of microorganisms [17,18]. The calculation formula is:VL = SQRT{[lnβG:ln(NAG + LAP)]^2^ + (lnβG:lnAP)^2^}
VA = Degrees{ATAN2[lnβG:lnAP,lnβG:ln(NAG + LAP)]}
where a longer VL means a greater relative C limit of microorganism; VA < 45°or >45° indicates the relative degree of N or P restriction, respectively.

### 4.5. Statistical Analysis

The experimental data were sorted by Excel 2019 (Microsoftware 2019), plotted by origin 2021 software, and analyzed by Pearson correlation. SPSS 19.0 was used to analyze the variance of one factor and two factors, and the least significant difference method was used to test the difference. Furthermore, redundancy analysis (RDA) was used after the enzyme activity data underwent Hellinger transformation and environmental factor data were standardized in an effort to determine the most significant factors that shaped the soil ecoenzyme activities and ecoenzymatic stoichiometry using the Vegan package in R.

## 5. Conclusions

With stoichiometric of soil C:N:P acquisition ratios measured by ln(BG):ln(LAP + NAG):ln(AP) activities in our experimental area were close to 1:1:1. Film mulching measures can significantly improve the soil extracellular enzyme activity in different growth stages, which is more conducive to maintaining the balance of soil nutrients. Our study also proves that soil in Loess Plateau was limited by N and P, and when the nitrogen application levels were less than 225 kg·hm^−2^, the soil enzyme activity was mainly limited by N nutrient, and when the nitrogen application level was 300 kg·hm^−2^, it was mainly limited by P nutrient. When the nitrogen application level was 225 kg·hm^−2^, the soil enzyme activity was the highest, and the metering ratio was the closest to 1:1:1. Our research revealed that the ratios of soil C:P, C:N, N:P and pH were the main environmental factors influencing the soil enzyme activity and stoichiometric ratio. These findings may be crucial for us to understand microbial metabolic limitations and nutrient cycles in arid and oligotrophic ecosystems. Our study provides comprehensive evidence for the response of soil EEA to different levels of nitrogen and the relationship between them and soil carbon, nitrogen and phosphorus under plastic film mulching in dry farmland of northern China. Considering the limitation of single soil enzyme activity research, we expect further research on microbial biomass (for example, PLFAs and functional gene abundance, etc.) in future research in order to reveal the relationship between soil enzyme activity and microbial biomass in dry farmland.

## Figures and Tables

**Figure 1 plants-11-01748-f001:**
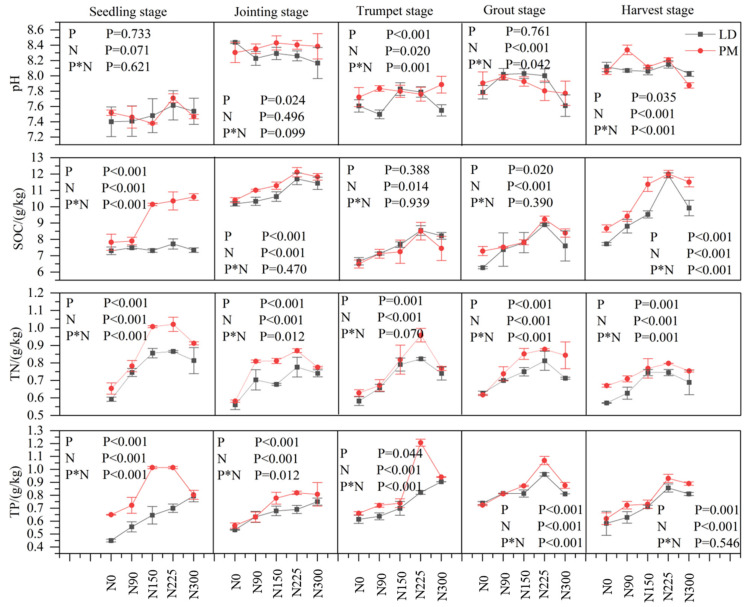
Physical and chemical properties of soil under the different treatments. SOC, soil organic carbon; TN, total carbon; TP, total phosphorus; P, plastic film mulching; N, nitrogen levels (0, 90, 150, 225, and 300 kg·hm^−2^); P*N, plastic film mulching * nitrogen levels; N0, nitrogen application level was 0 kg·hm^−2^; N90, nitrogen application level was 90 kg·hm^−2^; N150, nitrogen application level was 150 kg·hm^−2^; N225, nitrogen application level was 225 kg·hm^−2^; N300, nitrogen application level was 300 kg·hm^−2^.

**Figure 2 plants-11-01748-f002:**
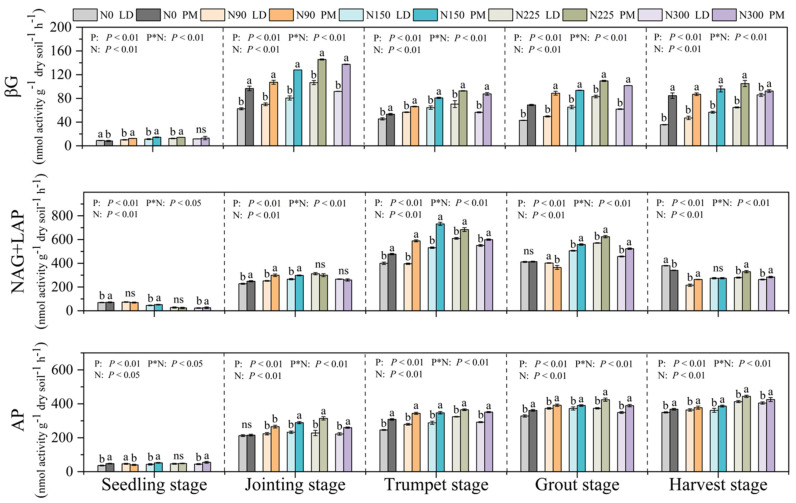
Soil enzyme activities under different treatments. P, plastic film mulching; N, nitrogen levels; P*N, plastic film mulching * nitrogen levels; N0, nitrogen application level was 0 kg·hm^−2^; N90, nitrogen application level was 90 kg·hm^−2^; N150, nitrogen application level was 150 kg·hm^−2^; N225, nitrogen application level was 225 kg·hm^−2^; N300, nitrogen application level was 300 kg·hm^−2^. Different lowercase letters (a, b) indicate that there was a significant difference between the PM and LD treatments at the same nitrogen level, and “ns” indicates that there was no significant difference between the PM and LD treatments at the same nitrogen level.

**Figure 3 plants-11-01748-f003:**
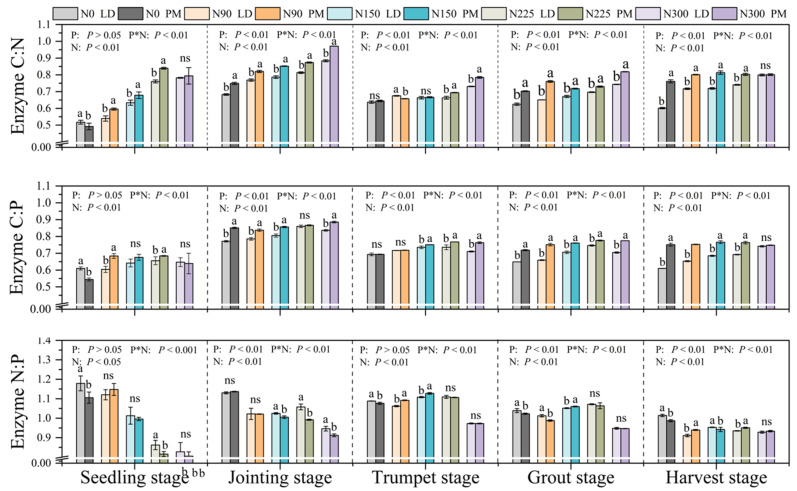
The stoichiometric ratio of soil enzyme activities under different nitrogen application levels under PM and LD treatments. P, plastic film mulching; N, nitrogen levels; P*N, plastic film mulching * nitrogen levels; N0, nitrogen application level was 0 kg·hm^−2^; N90, nitrogen application level was 90 kg·hm^−2^; N150, nitrogen application level was 150 kg·hm^−2^; N225, nitrogen application level was 225 kg·hm^−2^; N300, nitrogen application level was 300 kg·hm^−2^. Different lowercase letters (a, b) indicate that there was significant difference between the PM and LD treatments at the same nitrogen level, and “ns” indicates that there was no significant difference between the PM and LD treatments at the same nitrogen level.

**Figure 4 plants-11-01748-f004:**
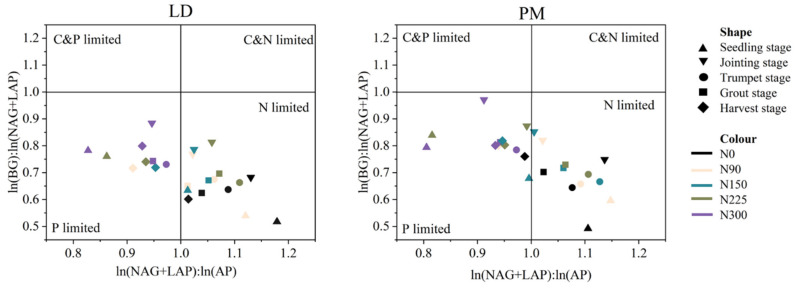
Stoichiometric ratio of soil enzyme activities at different nitrogen levels under the PM and LD treatments. C, soil organic carbon; N, total carbon; P, total phosphorus; N0, nitrogen application level was 0 kg·hm^−2^; N90, nitrogen application level was 90 kg·hm^−2^; N150, nitrogen application level was 150 kg·hm^−2^; N225, nitrogen application level was 225 kg·hm^−2^; N300, nitrogen application level was 300 kg·hm^−2^.

**Figure 5 plants-11-01748-f005:**
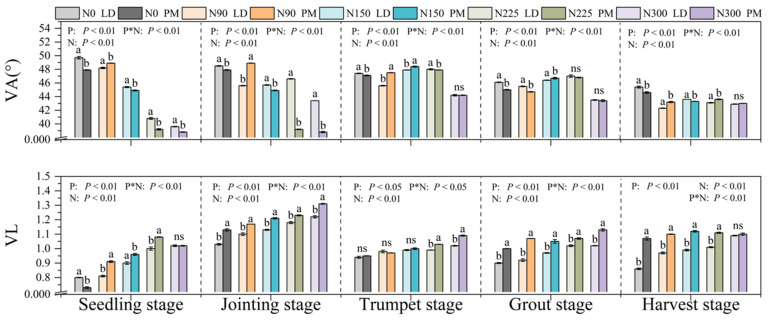
Vector analysis of soil enzymes under the different treatments. VA, vector angel (°); VL, vector length; P, plastic film mulching; N, nitrogen levels; N*P, plastic film mulching * nitrogen levels; N0, nitrogen application level was 0 kg·hm^−2^; N90, nitrogen application level was 90 kg·hm^−2^; N150, nitrogen application level was 150 kg·hm^−2^; N225, nitrogen application level was 225 kg·hm^−2^; N300, nitrogen application level was 300 kg·hm^−2^. Different lowercase letters (a, b) indicate that there was a significant difference between the PM and LD treatments at the same nitrogen level, and “ns” indicates that there was no significant difference between PM and LD treatments at the same nitrogen level.

**Figure 6 plants-11-01748-f006:**
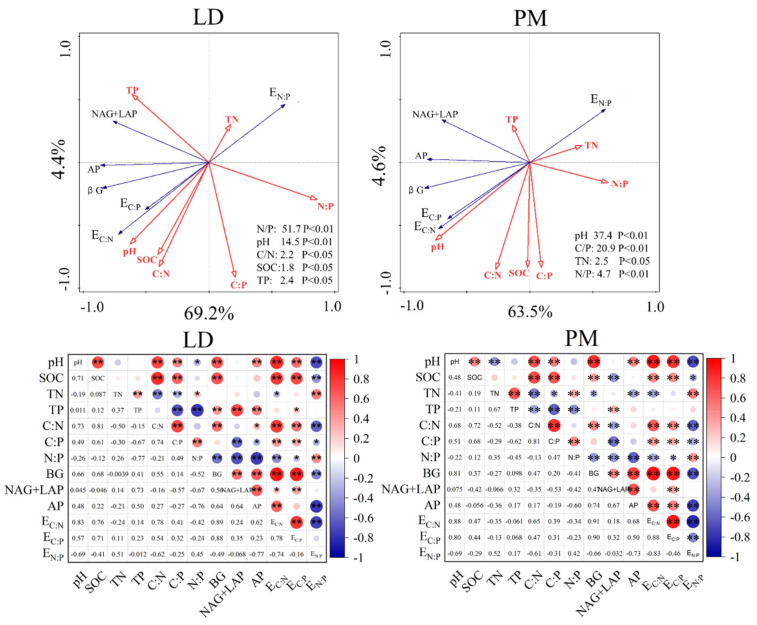
Redundancy analysis of soil physical and chemical properties and soil enzyme activities under different nitrogen application levels of PM and LD treatments. * *p* < 0.05, ** *p* < 0.01.PM, plastic film mulching; LD, no film mulching.

**Table 1 plants-11-01748-t001:** Significance of the effects of different treatments.

	pH	SOC	TN	TP
S	<0.001 **	<0.001 **	<0.001 **	<0.001 **
P	=0.001 **	<0.001 **	<0.001 **	<0.001 **
N	<0.001 **	<0.001 **	<0.001 **	<0.001 **
S*P	=0.028 *	<0.001 **	<0.001 **	<0.001 **
S*N	<0.001 **	<0.001 **	<0.001 **	<0.001 **
P*N	=0.023 *	=0.042 *	<0.001 **	<0.001 **

S, maize growth stage (seeding stage, jointing stage, trumpet stage, grout stage and harvest stage); P, plastic film mulching; N, nitrogen levels (0, 90, 150, 225 and 300 kg·hm^−2^); SOC, soil organic carbon; TN, total carbon; TP, total phosphorus. ** Indicates *p* < 0.01, extremely significantly different; * indicates *p* < 0.05, significantly different.

## Data Availability

The data presented in this study are available from the corresponding author upon request.

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
