# Peer review of "Effects of Plastic Film Mulching on Soil Enzyme Activities and Stoichiometry in Dryland Agroecosystems"

_plants, 2022, doi:10.3390/plants11131748_

Round 1

Reviewer 1 Report

The subject of the manuscript is consistent with the scope of the Journal. The authors should make an effort to point out specificity of this study, and what is its continuation to the scientific existing knowledge.

Please better explain why these studies were taken, what new they would bring to science.

Please check that the enzyme activity is definitely well calculated, eg the activity of β-glucosidase (βG) and alkaline phosphatase is extremely high.

Please, be sure that all the references cited in the manuscript are also included in the reference list and vice versa with matching spellings and dates.

Additional comments from the reviewer:

In the Materials and Methods section, please describe in detail how the enzyme activity was calculated as it is very high. It would also be nice to present a pattern. Please also provide the procedure for the determination of soil organic carbon, total nitrogen and total phosphorus. Give the apparatus and standards.

In the Introduction chapter, please present the research hypotheses. 

Author Response

Dear reviewers,

We are very grateful to you for giving us an opportunity to revise our manuscript. we appreciate you very much for your positive and constructive comments and suggestions on our manuscript entitled “Effects of plastic film mulching on soil enzyme activities and stoichiometry in dryland agroecosystems” (ID: plants-1774072)

We have studied reviewers’ comments carefully and tied our best to revise our manuscript according to the comments. The following are the responses and revisions I have made in responses to the reviewers’ questions and suggestions on an item-by-item basis. Thanks again to the hard work of the editor and reviewer!

1. The subject of the manuscript is consistent with the scope of the Journal. The authors should make an effort to point out specificity of this study, and what is its continuation to the scientific existing knowledge.

Response 1: We greatly appreciate the constructive comments and suggestions. we which have been revised in detail in the Introduction section of the paper. The modified parts have been marked in red font.

2. Please better explain why these studies were taken, what new they would bring to science.

Response 2: Thank you for your suggestion, the reasons why we carry out this research are as follows: Firstly, a large number of studies have proved that soil EEA is often used as a powerful index to evaluate soil quality and microbial metabolic activity because of its functional characteristics and sensitivity to environmental changes. Secondly, as the main technical means to increase agricultural production, there is little research on the response mechanism of soil EEA in the whole growth period under the comparison of mulching and non-mulching. Finally, optimizing the application level of nitrogen fertilizer under plastic film and improving the utilization efficiency of nitrogen fertilizer are of great significance to realize the sustainable development of China's agricultural environment. This study provides a comprehensive evidence for the response of soil enzyme activities and stoichiometry to different levels of nitrogen and the relationship between them and soil carbon, nitrogen and phosphorus under plastic film mulching in dryland agroecosystems of northern China.

we which have been revised in detail in the Introduction section of the paper. The modified parts have been marked in red font.

3. Please check that the enzyme activity is definitely well calculated, eg the activity of β-glucosidase (βG) and alkaline phosphatase is extremely high.

Response 3: Thank the reviewers for your corrections. After inspection, there is no error in calculation, and there is an error in the unit. See paper 4.4 for specific calculation steps.

4. Please, be sure that all the references cited in the manuscript are also included in the reference list and vice versa with matching spellings and dates.

Response 4: Thank you for your suggestion, we have checked and revised the text in detail.

Additional comments from the reviewer:

5. In the Materials and Methods section, please describe in detail how the enzyme activity was calculated as it is very high. It would also be nice to present a pattern. Please also provide the procedure for the determination of soil organic carbon, total nitrogen and total phosphorus. Give the apparatus and standards.

Response 5: Think you for your suggestions, we have made detailed revisions in 4. materials and methods. The modified parts have been marked in red font.

6. In the Introduction chapter, please present the research hypotheses. 

Response 6: Thank you for your suggestion, I have made detailed revisions in the Introduction. The modified parts have been marked in red font.

Once again, thank you very much for your comments and suggestions. 

Reviewer 2 Report

The article of Liu et al. entitled “Effects of plastic film mulching on soil enzyme activities and stoichiometry in dryland agroecosystems” presents an interesting approach in terms of understanding the effect of plastic film mulching on soil enzyme activities under different N fertilization rates. The experimental design is well-prepared, the null hypotheses well-presented and the dataset in the ms quite interesting. However, I have some concerns in relation to the content of this work, in terms of making several speculations on relating the microbial activity to the enzyme activity. I would have expected the authors to present some microbial analysis data (e.g. PLFAs) to support further their results. Presenting only the results of enzymes does not make the results very solid, a lot of speculation is included and thus we can not have safe conclusions. This is the novel part of this study but, in my opinion, presenting data related only to enzyme activity is not sufficient enough to support the publication of this article.

Other comments

1.     The text must be edited by an English native speaker, as many grammar issues can be found in the text

2.     The authors must pay attention to the statistical analysis that they applied to their dataset and how they decided to present them. They applied factorial ANOVA in Figures 1, 2, and 3 which is the correct approach, but in Table 2 we see a t-Test, and the way that they present the letters is very complicated. Additionally, in figures 2 and 3, the reader would expect to see letters above the columns to indicate the significant statistical differences. Also, Table 1 presents the results including time, while in Figure 1 we see the different stages. This is also quite confusing, especially since the statistical values differ. The authors must explain why they decided to present the same dataset in two different ways in their paper.

3.     In the statistical analysis section of the M&M, the authors must also include the RDA analysis and also to explain whether they had to transform their data prior to statistical analysis

4.     The discussion is very long and several descriptive parts should be omitted.

5.     Please try to have self-sufficient figure legends. All abbreviations mentioned there must be explained. 

     The quality of the figures must also be improved

6.     Lines 94-108. Please try to re-write this part. It is very descriptive and in quite confusing. There is no need to present so many details.

T

Author Response

Dear reviewers,

We are very grateful to you for giving us an opportunity to revise our manuscript. we appreciate you very much for your positive and constructive comments and suggestions on our manuscript entitled “Effects of plastic film mulching on soil enzyme activities and stoichiometry in dryland agroecosystems” (ID: plants-1774072 )

We have studied reviewers’ comments carefully and tied our best to revise our manuscript according to the comments. The following are the responses and revisions I have made in responses to the reviewers’ questions and suggestions on an item-by-item basis. Thanks again to the hard work of the editor and reviewer!

1. The article of Liu et al. entitled “Effects of plastic film mulching on soil enzyme activities and stoichiometry in dryland agroecosystems” presents an interesting approach in terms of understanding the effect of plastic film mulching on soil enzyme activities under different N fertilization rates. The experimental design is well-prepared, the null hypotheses well-presented and the dataset in the ms quite interesting. However, I have some concerns in relation to the content of this work, in terms of making several speculations on relating the microbial activity to the enzyme activity. I would have expected the authors to present some microbial analysis data (e.g. PLFAs) to support further their results. Presenting only the results of enzymes does not make the results very solid, a lot of speculation is included and thus we can not have safe conclusions. This is the novel part of this study but, in my opinion, presenting data related only to enzyme activity is not sufficient enough to support the publication of this article.

Thank you for your query. Our paper focuses on the relationship between enzyme activity and soil nutrients, part of which is the input of nutrients and part of which is the relationship with soil nutrients. At first, we thought that soil enzyme activity mainly came from soil microbes, but it really didn't represent microbes, so the speculation about soil microbes has been deleted in this paper. After careful thinking and discussion, we agreed that what the experts said is very important. We will supplement the indicators such as microbial indicators PLFA in the next experiment. This has been supplemented in the conclusions. Thank you very much for your suggestion.

Other comments

2. The text must be edited by an English native speaker, as many grammar issues can be found in the text

Response 2: We apologize for the poor language of our manuscript. We have now worked on both language and readability and have also involved native English speakers for language corrections.

3. The authors must pay attention to the statistical analysis that they applied to their dataset and how they decided to present them. They applied factorial ANOVA in Figures 1, 2, and 3 which is the correct approach, but in Table 2 we see a t-Test, and the way that they present the letters is very complicated. Additionally, in figures 2 and 3, the reader would expect to see letters above the columns to indicate the significant statistical differences. Also, Table 1 presents the results including time, while in Figure 1 we see the different stages. This is also quite confusing, especially since the statistical values differ. The authors must explain why they decided to present the same dataset in two different ways in their paper.

Response 3: We have modified Table 2 (now changed to Figure 5) and marked Figures 2, 3 and 5 with letters. The time in table 1 represents the different stage, which has been revised. Thank you for your advice.

4. In the statistical analysis section of the M&M, the authors must also include the RDA analysis and also to explain whether they had to transform their data prior to statistical analysis.

Response 4: Thanks to the reviewers for your suggestions, the contents are now added as follows:“Furthermore, redundancy analysis (RDA) were used after the enzyme activity data underwent Hellinger transformation and environmental factor data was standardized, in an effort to determine the most significant factors that shaped soil ecoenzyme activities and ecoenzymatic stoichiometry using the Vegan package in R.”The modified parts are marked in red font in the text.

5. The discussion is very long and several descriptive parts should be omitted.

Response 5: Thanks to the comments made by the reviewers, we have made appropriate cuts to the discussion section.

6. Please try to have self-sufficient figure legends. All abbreviations mentioned there must be explained. The quality of the figures must also be improved

Response 6: I am very grateful to the reviewers for your comments, and have made appropriate modifications to the charts in this article.

7. Lines 94-108. Please try to re-write this part. It is very descriptive and in quite confusing. There is no need to present so many details.

Response 7: Thank you for your suggestions. The modified parts are marked in red font in the text.

Once again, thank you very much for your comments and suggestions. 

Reviewer 3 Report

1

1.     The relevance, importance and innovative aspects of the paper should be more evident.

2.     Words used in the title should not be repeated in the keywords.

3.     Why the analyzes concern only one year of research?

4.     The methodology (Soil extracellular enzyme activity and enzyme stoichiometry ratio)  should be described in more detail.

5.     Figures no. 2 and 5 are not legible.

6.     Add your recommendations for future research.

7.     Make sure the references are added correctly according to the journal's instructions.

8.     The language correctness should be verified by a native speaker.

1    

Author Response

Dear reviewers,

We are very grateful to you for giving us an opportunity to revise our manuscript. we appreciate you very much for your positive and constructive comments and suggestions on our manuscript entitled “Effects of plastic film mulching on soil enzyme activities and stoichiometry in dryland agroecosystems” (ID: plants-1774072)

We have studied reviewers’ comments carefully and tied our best to revise our manuscript according to the comments. The following are the responses and revisions I have made in responses to the reviewers’ questions and suggestions on an item-by-item basis. Thanks again to the hard work of the editor and reviewer!

1. The relevance, importance and innovative aspects of the paper should be more evident.

Response 1: Thank the reviewers for your valuable comment, which have been revised in detail in the Introduction section of the paper. The modified parts have been marked in red font.

2. Words used in the title should not be repeated in the keywords.

Response 2: Thank you for your suggestion. The corresponding modifications have been made in the paper.

3. Why the analyzes concern only one year of research?

Response 3: Thank the reviewers for your question. Our explanation is as follows: the first experiment started in 2015, and each treatment was repeated three times. Up to now, seven years of field planting experiments have been carried out, and the soil environmental conditions and microbial conditions are basically stable. At the same time, we sampled five times before and after the whole growth period of maize in 2021, which was enough to explain the response mechanism of soil EEA to film mulching and nitrogen application levels.

4. The methodology (Soil extracellular enzyme activity and enzyme stoichiometry ratio) should be described in more detail

Response 4: Thank you for your suggestion, which have been revised in detail in 4.4 materials and methods of the paper. The modified parts have been marked in red font.

5. Figures no. 2 and 5 are not legible.

Response 5: Thank the reviewers for your valuable comments. The corresponding modifications have been made in the paper.

6. Add your recommendations for future research.

Thank the reviewers for your suggestion, which have been supplemented in the conclusion part of the paper, and the relevant parts are marked in red font.

7. Make sure the references are added correctly according to the journal's instructions.

Response 7: Thanks to reviewers for your rigorous attitude. This article has been carefully revised to reflect your comments.

8. The language correctness should be verified by a native speaker.

Response 8: We apologize for the poor language of our manuscript. We have now worked on both language and readability and have also involved native English speakers for language corrections.

Once again, thank you very much for your comments and suggestions. 

Round 2

Reviewer 1 Report

The submitted version of the manuscript is much better than the previous one. I appreciate the authors' corrections. In my opinion, the manuscript is suitable for publication.

Reviewer 2 Report

This version is significantly improved compared to the previous one. The authors have addressed my comments and I suggest the publication of the article. Nevertheless, some changes need to be made.

Lines 425-429. Please try using the same letters as those found in the equation. Do not use capital letters (e.g. A) to describe a factor in the equation "a".

I still believe that several linguistic issues exist in the text. Among others:

Line 385. "a use spectrophotometer" omit use

line 450. change "The stoichiometric" with stoichiometry